# Using Synthetic Data to Improve the Long-range Forecasting of Time Series Data

## Abstract

Effective long-range forecasting of time series data remains an unsolved and open problem. One possible approach is to use generative models to improve long-range forecasting, but the challenge then is how to generate high-quality synthetic data. In this paper, we propose a conditional Wasserstein GAN with Gradient and Error Penalty (cWGAN-GEP), aiming to generate accurate synthetic data that preserves the temporal dynamics between the conditioning input and generated data. By using such synthetic data, we develop a long-range forecasting method called Generative Forecasting (GenF). GenF consists of three key components: (i) a cWGAN-GEP based generator, to generate synthetic data for next few time steps. (ii) a predictor which makes long-range predictions based on generated and observed data. (iii) an information theoretic clustering (ITC) algorithm to better train the cWGAN-GEP based generator and the predictor. Our experimental results on three public datasets demonstrate that GenF significantly outperforms a diverse range of state-of-the-art benchmarks and classical approaches. In most cases, we find an improvement of at least 10% over all studied methods. Lastly, we conduct an ablation study to demonstrate the effectiveness of the cWGAN-GEP and the ITC algorithm.

## 1 Introduction

Short-range forecasting of time series data has been able to provide some useful information, but its scope of application is limited (Chatfield, 2000; Granger & Newbold, 2014). In most applications, long-range forecasting of time series data is preferred as it allows more time for early intervention and planning opportunities (Alvarez et al., 2010; Azad et al., 2014). For example, long-range forecasting of patient's vital signs effectively gives clinicians more time to take action and may reduce the occurrence of potential adverse events (Luo, 2015; Choi & et al, 2016).

However, a critical issue affecting the long-range forecasting is that the predictive performance (e.g., $N$-step ahead) becomes worse as $N$ grows. One practical approach to address this problem is to use synthetic data to shorten the prediction horizon $N$. For example, in iterative forecasting (Marcellino et al., 2006; Hamzaçebi et al., 2009), the previous predictions are used together with the original data as new inputs to evaluate the next prediction. However, the synthetic data (i.e., previous predictions) generated in this recursive and supervised manner is more susceptible to error propagation (Sorjamaa & et al, 2007; Weigend & Gershenfeld, 2018). Overall, the quality of synthetic data is the key to improving long-range forecasting. In recent years, the success of Generative Adversarial Network (GAN) (Goodfellow & et al, 2014) in replicating real-world content has inspired numerous GAN based architectures (Frid-Adar et al., 2018a;b; Shin et al., 2018; Bowles et al., 2018), in order to generate synthetic data for different purposes (e.g., improve classification accuracy). However, the utilization of synthetic data to improve long-range forecasting remains unexplored.

In this paper, we contribute to the area of long-range forecasting as follows.

1. We augment the existing conditional Wasserstein GAN with Gradient Penalty with a mean squared error term. This new architecture, called cWGAN-GEP, aims to generate accurate synthetic data which preserves the temporal dynamics between the conditioning input and generated data.
2. We develop a long-range forecasting method called Generative Forecasting (GenF) which consists of three key components: (i) a cWGAN-GEP based generator, to generate synthetic data for next few time steps. (ii) a predictor which makes long-range predictions based on generated and observed data. (iii) an information theoretic clustering (ITC) algorithm to better train the cWGAN-GEP based generator and the predictor.

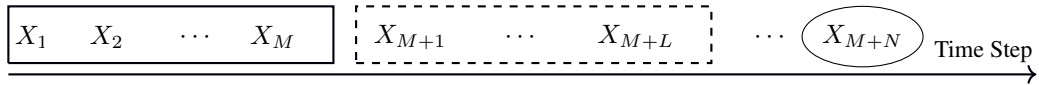

Observation Window (Length = $M$)   Synthetic Data Window (Length = $L$)   Prediction Horizon (Length=$N$)

Figure 1: Long-range forecasting via observation/synthetic data window and prediction horizon.

3. We conduct experiments on three public time series datasets and our results demonstrate that GenF significantly outperforms a diverse range of state-of-the-art benchmarks and classical approaches. In most cases, we find improvements of at least 10% over the benchmark methods.

To the best of our knowledge, our work is the first to propose using synthetic data generated by a GAN based architecture to improve long-range forecasting and the proposed cWGAN-GEP is not explored in any of the previous work.

## 2 BACKGROUND ON TIME SERIES FORECASTING

In time series forecasting, we use an observation window of historical data to predict the future value (see Fig. 1). Let $M$ denote the observation window length and $N$ denote the prediction horizon. For those methods that contain generative models, let $L$ denote the synthetic data window length. Let $X_i \in \mathbb{R}^K$ be the observation at the $i_{th}$ time step, where $K$ is the number of features. In Fig. 1, the prediction horizon is $N$, indicating that we plan to make predictions $N$ time steps ahead, i.e., at time step $t + N$ (e.g., the circled entry in Fig. 1). Next, we discuss related work in Section 2.1 and shortlist several benchmark methods for comparison in Section 2.2.

### 2.1 RELATED WORK

Early methods which use neural networks to perform long-range forecasting include (Nguyen & Chan, 2004; Goswami & Srividya, 1996; Al-Saba & El-Amin, 1999). For example, Nguyen & Chan (2004) propose a group of neural networks to make predictions at different time steps. Several recent works (Yu et al., 2017; Lai et al., 2018; Qiu et al., 2020; Bouktif et al., 2018; Barsoum et al., 2018) attempt to improve the long-range forecasting by proposing new architectures. For example, Yu et al. (2017) propose a Long Short-Term Memory (LSTM) (Hochreiter & Schmidhuber, 1997) based Tensor-Train Recurrent Neural Network as a module for sequence-to-sequence (Sutskever et al., 2014) framework, called TLSTM. Moreover, Lai et al. (2018) propose a Long- and Short-term Time-series network (LSTNet) which combines LSTM with autoregressive models to minimize a customized loss function inspired by support vector regression (SVR) (Cortes & Vapnik, 1995). Another recent work (Cheng et al., 2020) proposes Multi-Level Construal Neural Network (MLCNN) which attempts to improve the predictive performance by fusing forecasting information of different future time. One more interesting work to mention is (Ahmed et al., 2007), which combines Gaussian processes (Rasmussen, 2003) and neural networks to improve the forecasting performance. Overall, all these methods can be classified into two main classes: direct forecasting and iterative forecasting (Hamzaçebi et al., 2009; Bontempi et al., 2012).

In direct forecasting, a group of models are trained to directly make predictions at different values of $N$. The advantage is that all models are independent of each other and hence, models do not interfere with each other. However, its performance tends to becomes worse with $N$ due to the lack of long-range dependencies. In iterative forecasting, the model is trained to make predictions for the next time step ($t + 1$) only. Then, the same model will be used over and over again and previous predictions are used together with the past observations to make predictions for the next time step. This process is recursively repeated $N$ times to make predictions $N$ time steps ahead. The previous predictions can be considered as synthetic data (with synthetic data window length $L = N - 1$) to shorten the effective prediction horizon. However, the synthetic data generated in this supervised and recursive manner is more susceptible to error propagation, meaning that a small error in the current prediction becomes larger error in subsequent predictions (Taieb et al., 2012; Sorjamaa & et al, 2007).

### 2.2 BENCHMARK METHODS

The taxonomy in Section 2.1 suggests that most existing methods can be classified as either direct or iterative forecasting. This motivates us to use both direct and iterative forecasting as benchmark methods. As mentioned in Section 2.1, some classical models such as LSTM, Autoregressive Integrated Moving Average (ARIMA) (Box & Pierce, 1970), SVR, and Gaussian Process Regression (GPR) (Rasmussen, 2003) are used to form the core of many state-of-the-art methods. Hence, we

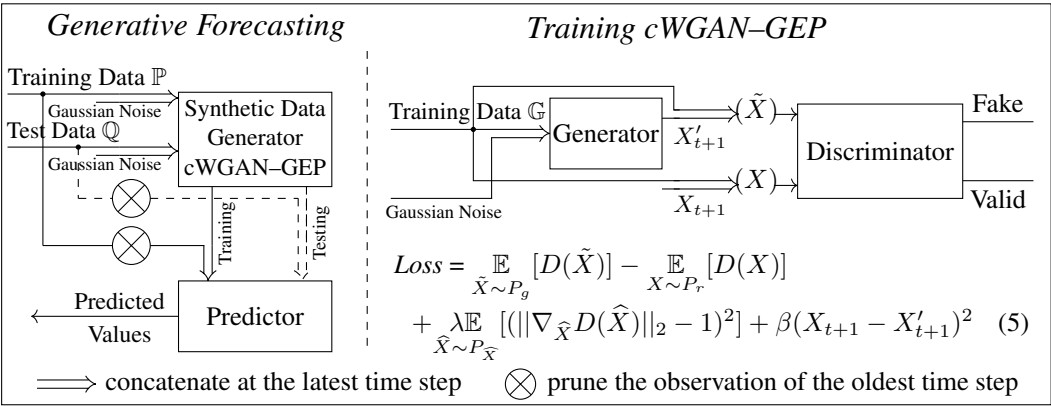

Figure 2: (left) cWGAN-GEP generates synthetic data for the next time step of training data $\mathbb{P}$ and test data $\mathbb{Q}$. The modified training data $\mathbb{P}$ is used to train the predictor to perform predictions for the modified test data $\mathbb{Q}$. (right) Training of cWGAN-GEP via training data $\mathbb{G}$.

use these four classical models to implement both direct and iterative forecasting. In addition, three strong baselines described in Section 2.1, i.e., TLSTM, LSTNet, and MLCNN, are also selected for comparison as they are reported to provide outstanding long-range forecasting performance in the literature (Yu et al., 2017; Lai et al., 2018; Cheng et al., 2020). Moreover, the source code released for these works is well organized to ensure correct implementation.

# 3 GENERATIVE FORECASTING

In this section, we introduce the idea of GenF (see Fig. 2) and describe its three components: cWGAN-GEP based synthetic data generator, LSTM based predictor and the ITC algorithm.

## 3.1 IDEA OF GENERATIVE FORECASTING

To improve long-range forecasting, we develop an approach called Generative Forecasting (GenF) which consists of a cWGAN-GEP based generator, to generate synthetic data for next few time steps, and a predictor, to make long-range predictions based on observed and generated data. Assuming we use past 3 observations $\{X_t, X_{t-1}, X_{t-2}\}$ and make predictions at $t + 10$, we can express GenF as

$$\text{Generate synthetic data} \begin{cases} X'_{t+1} = g(X_t, X_{t-1}, X_{t-2}, \theta_g), & (1) \\ X'_{t+2} = g(X'_{t+1}, X_t, X_{t-1}, \theta_g), & (2) \\ X'_{t+3} = g(X'_{t+2}, X'_{t+1}, X_t, \theta_g), & (3) \end{cases}$$

$$\text{Make a long-range prediction:} \quad \bar{X}_{t+10} = f_{10}(X'_{t+3}, X'_{t+2}, X'_{t+1}, \theta_{10}), \quad (4)$$

where $g$ denotes the cWGAN-GEP based generator with parameter $\theta_g$ and $f_{10}$ is the predictor with parameter $\theta_{10}$. Let $X'_{t+1}$, $X'_{t+2}$ and $X'_{t+3}$ denote the generated synthetic data for next three time steps and $\bar{X}_{10}$ denote the prediction at $t + 10$. Note that the value of the synthetic data window length $L$ depends on the quality of the synthetic data and (1) - (3) is an example of $L = 3$. It can be seen that the generator $g$ generates synthetic data via the strategy of iterative forecasting (see (1) - (3)) and the predictor makes a prediction via the strategy of direct forecasting (see (4)). We note that GenF does not belong to either direct or iterative forecasting. The key difference is that GenF leverages synthetic data to shorten the effective prediction horizon while direct forecasting does not leverage any synthetic data. Moreover, unlike iterative forecasting where the synthetic data window length is $L = N - 1$, the synthetic data window length $L$ of GenF is much smaller than the prediction horizon $N$ ($L = 3$ and $N = 10$ in the example of (1) - (4)), so that it can prevent the propagation of error at an early stage. In such a manner, we argue that GenF attempts to balance direct forecasting and iterative forecasting.

## 3.2 GENERATIVE FORECASTING: THE CWGAN-GEP BASED SYNTHETIC DATA GENERATOR

GAN (Goodfellow & et al, 2014) and its related works have demonstrated promising results in many generative tasks (Yi et al., 2017; Xu et al., 2017; Haidar & Rezagholizadeh, 2019; Zhang et al., 2017; Dong et al., 2018). The first GAN applied to time series data is C-RNN-GAN (Mogren, 2016) which

takes random noise as input and uses LSTM as both the generator and discriminator. One follow-up work, RCGAN (Esteban et al., 2017), makes use of the conditioning information to generate synthetic medical data. Along the way, many works have explored generating synthetic data in a diverse range of domains (e.g., sensor data (Alzantot et al., 2017), biosignal (Haradal et al., 2018)). We note that all these synthetic data are generated for some purpose (Dai & Le, 2015; Larsen et al., 2015; Dumoulin et al., 2016; Srivastava et al., 2015; Yoon et al., 2019). For example, (Frid-Adar et al., 2018a) makes use of synthetic data augmentation to improve the classification results, and (Bowles et al., 2018) mixes the synthetic data with training data to enhance the model's predictive ability. However, to the best of our knowledge, none of them have considered using the synthetic data generated by the GAN based architecture, to improve long-range forecasting.

We fill in this gap and propose a conditional Wasserstein GAN with Gradient and Error Penalty (Arjovsky et al., 2017; Mirza & Osindero, 2014; Gulrajani & et al, 2017) to generator synthetic data which can be used to improve long-range forecasting. The structure of cWGAN-GEP is shown in Fig. 2 (right). As can be seen, the training data $\mathbb{G}$ is used as a condition to direct the generation of synthetic data (Mirza & Osindero, 2014). Then, the synthetic data ($X'_{t+1}$) will be concatenated with the training data $\mathbb{G}$ at the latest time step, to be evaluated by the discriminator. The network configurations of cWGAN-GEP are given in Table 1 (left). The loss function is summarized as

$$Loss = \mathbb{E}_{\tilde{X} \sim P_g}[D(\tilde{X})] - \mathbb{E}_{X \sim P_r}[D(X)] + \lambda \mathbb{E}_{\widehat{X} \sim P_{\widehat{X}}}[(||\nabla_{\widehat{X}} D(\widehat{X})||_2 - 1)^2] + \beta(X_{t+1} - X'_{t+1})^2. \quad (5)$$

We note that the first two terms in the R.H.S of (5) correspond to Wasserstein distance which improves learning performance over the Jensen-Shannon divergence used in the original GAN (Arjovsky et al., 2017; Gulrajani & et al, 2017). The third term is the gradient penalty, corresponding to the 1-Lipschitz constraint. Unique to our work, we introduce the last term which attempts to minimize the mean squared error between the synthetic data and its true value. In such a manner, we argue that the generator can generate accurate synthetic data while preserving the temporal dynamics between conditioning input and generated data by minimizing the difference between the generated data distribution $P_g$ and real data distribution $P_r$.

### 3.3 GENERATIVE FORECASTING: THE LSTM BASED PREDICTOR

The LSTM unit (Hochreiter & Schmidhuber, 1997) is composed of a cell which remembers values over arbitrary time intervals and three gates regulate the flow of information into and out of the cell. Such a design can mitigate the issue of vanishing/exploding gradients and equips LSTM with the ability of learning and remembering over long sequence of inputs. In GenF, we use LSTM as the core of the predictor and the configuration of the predictor is given in Table 1 (left).

### 3.4 GENERATIVE FORECASTING: THE INFORMATION THEORETIC CLUSTERING ALGORITHM

We note that the datasets studied in this paper contain time series data for different patients, countries and sites (described in Section 4.1). In the following, we refer to patients, countries, sites as units. We note that the advantage of GenF is attempting to balance direct and iterative forecasting so as to improve the long-range forecasting performance. However, the price is that we need two independent datasets: datasets $\mathbb{G}$ and $\mathbb{P}$ to train the cWGAN-GEP based generator and the LSTM based predictor, respectively. It is possible to randomly split the entire training dataset at unit level into datasets $\mathbb{G}$ and $\mathbb{P}$, but the resulting datasets may not represent the entire training dataset well. In this subsection, we address this problem by suggesting an information theoretic clustering (ITC) algorithm.

Mutual Information (MI), denoted by $I(X; Y)$, is an idea from information theory (Cover & Thomas, 2006) that quantifies the dependency between random variables $X$ and $Y$. The higher the value of the MI, the better one variable can represent the other. The ITC algorithm consists of three steps: (i) assign a score to each unit by the scoring function $J(P_i) = \sum_{P_j \in \mathbb{D}, P_j \neq P_i} I(P_i, P_j)$, where $P_i$ refers to the candidate unit and $\mathbb{D} = \{P_1, P_2, \cdots\}$ is the dataset containing all units. To estimate MI, we use the KSG estimator (Kraskov et al., 2004) as all features studied are continuous variables (Gao et al., 2018; 2015). (ii) divide all units into $Z$ groups (of approximately equal size) based on the descending order of all scores, where $Z$ is a tunable parameter. Therefore, units with similar scores will be grouped together. Moreover, the units within the same group tend to be highly dependent on each other. (iii) randomly sample from each subgroup; this is equivalent to selecting typical units of each subgroup. Random proportional sampling from all groups gives a new training dataset $\mathbb{G}$ and the unsampled units form the new training dataset $\mathbb{P}$. In such manner, we can select more typical units to better train the cWGAN-GEP based generator and the LSTM based predictor in GenF.

| Generative Forecasting (GenF) | | | LSTNet |
| --- | --- | --- | --- |
| cWGAN-GEP | | Predictor | skip-length p = 5, gradient clipping = 10, |
| Generator | Discriminator | - | epoch = 1000, dropout = 0.1, batchsize = 64. |
| $(S_3, M, K)$ | $(S_3, (M{+}1), K)$ | $(S_1/S_2, M, K)$ | **TLSTM** |
| LSTM (5) | LSTM (5) | LSTM (5) | learning rate (lr) decay = 0.8, lr = $1e^{-3}$, |
| Linear (12) | Linear (12) | LSTM (5) | dropout = 0.1, batchsize = 64, epoch = 1000. |
| Linear (K) | Linear (4) | Linear (4) | **MLCNN** |
| Reshape $(S_3, 1, K)$ | Linear (1) | Linear (1) | span = 2, stride = 1, dropout = 0.2, gradient clipping = 10, epoch = 1000. |

Table 1: (left) Network configuration of Generative Forecasting. The fourth row represents the input variables and the parameter in Linear(·) is the number of output units in the linear layer. The parameter in LSTM(·) is the hidden state size. (right) Key parameters of the three selected baselines.

## 4 PERFORMANCE EVALUATION

In Section 4.1 and 4.2, we summarize the dataset information and described the experimental setup. In Section 4.3, we evaluate the performance of GenF and compare it to benchmark methods. Lastly, we conduct an ablation study to evaluate the efficiency of our framework design in Section 4.4.

### 4.1 DATASET DESCRIPTION

We select three public time series datasets: MIMIC-III Vital Signs dataset (Goldberger & et al, 2000), Multi-Site Air Quality dataset (Zhang & et al, 2017) and World Energy Consumption dataset (WorldBank, 2019). The MIMIC-III Vital Signs dataset contains the first 500 patients (from subject ID 23) in MIMIC-III Clinical database. For each patient, we extract 6 features: heart rate (bpm), respiratory rate, blood oxygen saturation (%), body temperature (°F), systolic and diastolic blood pressure (mmHg). Most (over 92%) vital signs are regularly recorded at a hourly interval over a duration of 144 hours on average. The Multi-Site Air Quality dataset includes air pollutants data from 12 monitoring sites. For each site, we extract the hourly record of PM10, $SO_2$, $NO_2$, $O_3$, PM2.5 and CO. We note that all features are in units of ug/m$^3$ and each site has 35,000 records on average. The World Energy Consumption dataset contains data from 128 countries and each country contains three annual energy consumption indicators: electricity (kWh per capita), fossil fuel (% of total) and renewable energy (% of total) from 1971 to 2014. For each dataset, a small amount of missing values are imputed using the last historical readings. Moreover, we scale all variables to [0,1] and reshape all scaled data via a sliding observation window into a three dimensional dataset $\mathbb{D} \in \mathbb{R}^{S \times M \times K}$.

### 4.2 EXPERIMENT SETUP

In the experiment, the dataset $\mathbb{D}$ is randomly split into three subsets at unit level: training dataset $\mathbb{T} \in \mathbb{R}^{S_1 \times M \times K}$ (60%), test dataset $\mathbb{Q} \in \mathbb{R}^{S_2 \times M \times K}$ (20%) and validation dataset $\in \mathbb{R}^{S_3 \times M \times K}$ (20%). We note that both GenF and the benchmark methods are trained using the training dataset $\mathbb{T}$, and the test dataset $\mathbb{Q}$ is used to evaluate the performance. To ensure **fair comparison**, we conduct grid search over all tunable hyper-parameters and possible configurations using the validation dataset. We highlight that all methods share the same grid search range and step size. Specifically, some key parameters are tuned as follows. (i) for all LSTM based networks (including GenF), the hidden state size of LSTM is tuned from 5 to 100 with step size of 5 and number of LSTM layers or dense layers are tuned from 1 to 10 with step size of 1. (ii) for SVR, the regularization coefficient $c$ is chosen from {0.1, 1, 10}. (iii) for GPR, the noise level $\alpha$ is chosen from {$1^{-10}$, $1^{-8}$, ..., 1, 10}.

We now show a list of tuned parameters for predicting heart rate using the MIMIC-III Vital Signs dataset. In terms of the four classical models (described in Section 2.2) used to implement iterative forecasting and direct forecasting, (i) SVR (kernel = rbf) and GPR (kernel = rbf) are trained using the flatted training dataset $\mathbb{T}$. (ii) ARIMA (2,0,1) is trained using the past $M$ historical values of a single feature (e.g., heart rate). (iii) for LSTM, we stack two LSTM layers (each LSTM with hidden size of 10) and two fully connected layers (with size of 10 and 1, respectively). This LSTM based neural network is directly trained using training dataset $\mathbb{T}$ for 1000 epochs using Adam (Kingma & Ba, 2014) with a learning rate of 0.001, in batches of 64. As for the three selected baselines (TLSTM, LSTNet, MLCNN), we refer to the source code released by their authors and some key parameters for those baselines are summarized in Table 1 (right). In terms of GenF, the training and testing

---

**Algorithm 1** Generative Forecasting Algorithm

---

**Require:** Time series dataset $\mathbb{D}$, the gradient penalty coefficient $\lambda=10$, the mean squared error penalty coefficient $\beta=15$, batch size $m = 64$, Adam Optimizer (lr = 0.001).
 1: Randomly split $\mathbb{D}$ at unit level: training dataset $\mathbb{T}$ (60%), test dataset $\mathbb{Q}$ (20%), validation dataset (20%).
 2: Apply the ITC algorithm to divide the training dataset $\mathbb{T}$ into two subsets at unit level: training dataset $\mathbb{G}$ (25%), training dataset $\mathbb{P}$ (35%).
 3: Training dataset $\mathbb{G} \xrightarrow{\text{train}}$ cWGAN-GEP based generator via loss function described in (5).
 4: **for** $i = 1$ to $L$ **do**
 5:     cWGAN-GEP $\xrightarrow{\text{generate}}$ synthetic data for the next time step of training dataset $\mathbb{P}$ & test dataset $\mathbb{Q}$.
 6:     Prune the observation of the oldest time step for training dataset $\mathbb{P}$ & test dataset $\mathbb{Q}$
 7:     Concatenate synthetic data at the latest time step of training dataset $\mathbb{P}$ & test dataset $\mathbb{Q}$ (see (1) - (3)).
 8: **end for**
 9: Modified training dataset $\mathbb{P} \xrightarrow{\text{train}}$ predictor, modified test dataset $\mathbb{Q} \xrightarrow{\text{test}}$ predictor.
10: **return** Mean squared error between predicted values and real values.

---

procedures are summarized in Algorithm 1. We note that both the cWGAN-GEP based synthetic data generator and the LSTM based predictor in GenF are trained using Adam (Kingma & Ba, 2014) with a learning rate of 0.001, in batches of 64 and sigmoid function. Furthermore, we note that the tuned parameters and configurations are similar for other datasets. Lastly, the source code will be released to enable reproducibility at the camera-ready stage.

We compare the performance of GenF to direct and iterative forecasting in Fig. 3. All results are averaged over 10 runs and the error-bars depict the standard deviation. The Mean Squared Error (MSE) between the predicted value and true value will be used to evaluate the predictive performance. In Fig. 3, we show three variants of GenF, namely GenF-1, GenF-2, GenF-3, where the numbers represent the value of synthetic data window length $L$. In some plots, we only show the performance of GenF-1 and GenF-2. This is because, in those plots, GenF-3 does not outperform GenF-1 or GenF-2, and, hence, is not shown. Lastly, the predictive performance for the time steps that are part of the generator are not shown. For example, GenF-1 indicates that we generate synthetic data for the next time step ($t + 1$) and make predictions for subsequent time steps ($t + 2, ...$). Since the value at $t + 1$ is part of the generator, the predictive performance at $t + 1$ is not shown.

### 4.3 Performance Comparison

In Fig. 3 (left column), we compare the performance of GenF to iterative/direct forecasting implemented by the four classical models. We find that the iterative/direct forecasting implemented by LSTM generally outperforms other implementations (e.g., GPR and SVR). This finding agrees with the result stated in (Ma et al., 2015; Lipton et al., 2015; Gers et al., 2002; Malhotra et al., 2015). For other models, ARIMA tends to work well on the small dataset (see Fig. 3 (e)). However, on larger datasets, ARIMA tends to perform worse (e.g., Fig. 3 (a), (g)) as it uses only a single feature which may not benefit from the interaction among multiple features. Some papers (Meyler et al., 1998; Contreras et al., 2003) also point out that ARIMA is poor at predicting turning points as it is essentially 'backward looking'. Moreover, we note that the performance of direct and iterative forecasting implemented by ARIMA are the same due to its recursive nature. Lastly, the performance of kernel based methods (GPR and SVR) generally lies in between LSTM and ARIMA (see Fig. 3 (a), (b)). We posit this is because the time series data can be non-stationary and the optimal kernel function may change over time (Wilson & Adams, 2013; Wilson et al., 2016).

In Fig. 3 (right column), we compare the performance of GenF to the three selected baselines. We note that, due to the design of MLCNN (i.e., leveraging the interaction between predictions), we only show its performance from $t + 3$ onwards. Moreover, the performance of GenF in the left and right columns looks different due to the different scales, but the values are the same. We find that the performance of the selected baselines generally can outperform iterative/direct forecasting and are comparable to some of the GenF variants. For example, the performance of MLCNN is comparable to GenF-1 when predicting fossil fuel consumption (see Fig. 3 (f)). In most cases, the performance of GenF is better than the selected baselines. For example, in Fig. 3 (b), the performance of MLCNN at $t + 4$ is 113.3 while the performance of GenF-3 at $t + 4$ is 103.2, which is approximately 10% lower. In other plots (see Fig. 3 (a), (h)), we can observe similar or greater improvements. Lastly, we have discussed with domain experts on the relevance of our long-range forecasting results (e.g., at $t + 4$) of the two vital signs. The feedback from clinicians is that the MSE performance is acceptable for interventional purposes for ICU patients and at-home high risk patients.

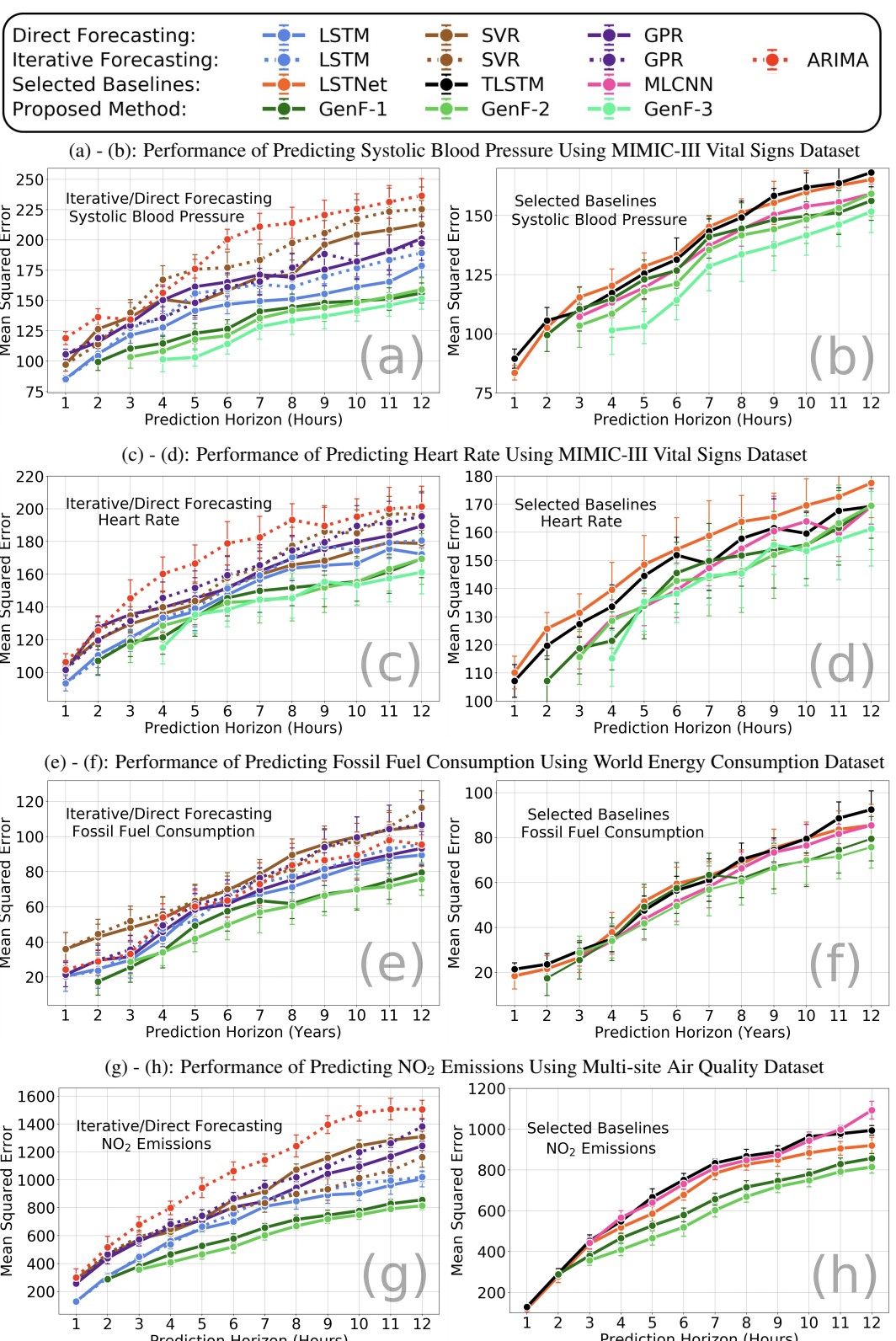

Figure 3: Performance comparison between three variants of GenF (GenF-1, GenF-2, GenF-3) and iterative/direct forecasting ((a), (c), (e), (g)), three selected baselines ((b), (d), (f), (h)). In each subfigure, we highlight the prediction target and the benchmark method used. Each plot is averaged over 10 runs and the error-bar is the standard derivation. Moreover, the performance of GenF in the left and right columns looks different due to the different scales, but the values are the same.

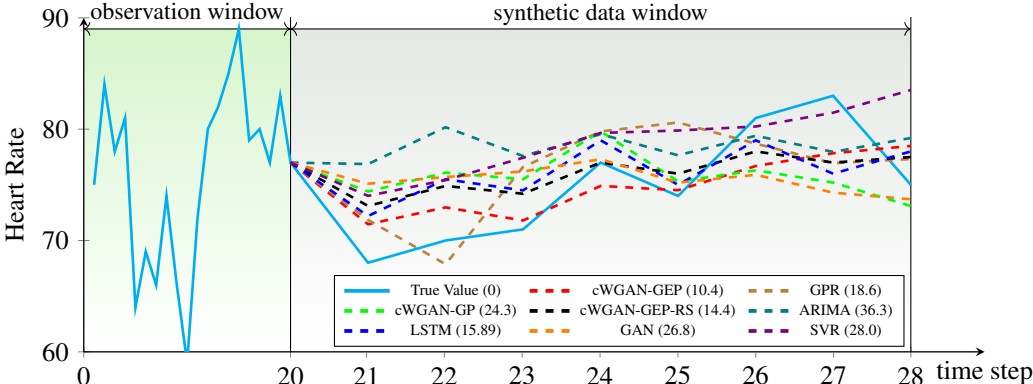

Figure 4: The synthetic data generated by cWGAN-GEP and its variants during the experiment of predicting heart rate using MIMIC-III Vital Signs dataset. As a comparison, we also show the true value (solid line) and the synthetic data generated by the iterative forecasting strategy via the four classical models. The value in parentheses is the mean squared error with respected to the true value.

## 4.4 ABLATION STUDY

To demonstrate the efficiency of several key components in GenF, we conduct an ablation study. Specifically, we remove one component at a time in GenF and observe the impact on performance. First, we construct several variants of cWGAN-GEP as follows.

• **cWGAN-GP:** The cWGAN-GEP without the mean squared error term in the loss function (i.e., without the 4th term on the right hand side of (5)).

• **cWGAN-RS:** The cWGAN-GEP without being trained using the ITC algorithm. Instead, the cWGAN-GEP is trained using a randomly selected subset of the entire training dataset $\mathbb{T}$.

• **GAN:** The original GAN (Goodfellow & et al, 2014) without considering the Wasserstein distance.

In the experiment of predicting heart rate, we evaluate the synthetic data generated by cWGAN-GEP and its three variants mentioned above. We shortlist an example of generating synthetic data for Subject ID 23 and show the past observations and the generated synthetic data in Fig. 4. Moreover, we also show the true value (solid line) and the synthetic data generated by the iterative forecasting strategy via the four classical models. The value in parentheses is the MSE relative to the true value.

For the synthetic data generated by the iterative forecasting strategy, LSTM has the smallest MSE which may explain why LSTM can outperform other classical models in long-range forecasting. Next, we compare the synthetic data generated by LSTM to cWGAN-GEP, which is essentially a combination of LSTMs (see Table 1(left)). We find that both LSTM and cWGAN-GEP can capture the rising trend of heart rate, but the synthetic data generated by cWGAN-GEP is more stable and accurate. Specifically, the performance of LSTM and cWGAN-GEP at time step = 21 (the first synthetic data point) are comparable. Subsequently, the synthetic data generated by LSTM tends to fluctuate greatly, resulting in a larger MSE. We posit this is caused by error propagation. As for cWGAN-GEP, the synthetic data is also generated recursively, but the synthetic data is more stable, leading to a smaller MSE. We posit this is the result of minimizing the difference in data distributions and similar stabilized effect can also be observed in (Zhu et al., 2017; Frid-Adar et al., 2018a).

When comparing cWGAN-GEP to cWGAN-GP, we find that removing the MSE term in (5) causes a significant drop in performance of synthetic data generation, suggesting the crucial role of the MSE term. Moreover, when comparing cWGAN-GEP to cWGAN-GEP-RS, the results demonstrate that the ITC algorithm has performance, suggesting its effectiveness in selecting typical units. As compared to the original GAN, cWGAN-GEP demonstrates superior performance in generating synthetic data as well. We posit this is because the cWGAN-GEP uses Wasserstein distance as part of the loss function, leading to a more stable learning process.

## 5 CONCLUSION

In this paper, we develop a method called Generative Forecasting which significantly outperforms benchmark methods in long-range forecasting. We point out that the success of GenF can be attributed to two components: cWGAN-GEP and the ITC algorithm, where the former can generate accurate and stable synthetic data, and the later helps to select more typical units for better training in GenF.

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
