# OpenReview forum: "Using Synthetic Data to Improve the Long-range Forecasting of Time Series Data"
_ICLR.cc/2021/Conference — Reject_

### Official Review · AnonReviewer1 · 2020-10-27

**Rating:** 5
**Confidence:** 4

**Review:**

Summary:
This paper proposes cWGAN-GEP for long-range forecasting of time-series data. cWGAN-GEP is a combination of data generation of data prediction, where given some observations, GAN iteratively generates a short synthetic time-series data, and an LSTM subsequently makes a long-range prediction based on the generated synthetic data. In order to train both components, the authors use information theoretic clustering. The proposed model outperforms various baselines on the prediction task using three time-series datasets.

Pros:
- The paper reads very well. The entire paper seems to be carefully written with as much details included as possible. Good presentation.
- The proposed method outperforms various baselines for time-series prediction using three different datasets.

Cons:
- It is not convincing to use a complicated GAN architecture for iterative data synthesis, when a simple LSTM could already do a decent job (as shown by figure 4). The authors should expand their ablation study by replacing the GAN module with an autoregressive LSTM/Transformer (with reasonably tuned hyperparameters) to evaluate the necessity of the complicated cWGAN-GEP.
- It is not well motivated why a two-stage approach (data synthesis first, and prediction later) is specifically helpful for long-range forecasting. The authors could have easily used an ensemble approach by using the two components to make predictions for all future time-steps and mixing (averaging or gating, etc) the predicted outcomes, which would be way easier since this will obviate the necessity of ITC. The authors should test this approach to justify the necessity of the two-stage approach.
- The claim that the proposed hybrid approach is helpful for long-range forecasting is not strongly justified. It seems that the performance gap between LSTM-based baseline models and the proposed methods stay rather unchanged as the prediction horizon widens. Maybe the proposed architecture is just generally a more powerful time-series predictor with many working parts.

After the author response:
Although the authors conducted additional experiments, they have only partially addressed the first comment regarding the use of LSTM instead of cWGAN-GEP by showing LSTM's comparison against cWGAN-GEP in synthetic data generation, when the question was to measure the forecasting performance, not synthetic data generation performance. Therefore increasing the score to 5.

---

> ### Author Response · Authors · 2020-11-19
> **Response to Reviewer 1**
>
> We would like to thank you for your valuable time and insightful comment. Please see our responses below.
>
> **1. On Model Complexity:**  **We note that cWGAN-GEP actually has less parameters than the LSTM baseline and thus is arguably simpler.** The LSTM baseline in Fig. 4 is carefully tuned and contains two LSTM layers and two fully connected layers, for a total of 3.5K parameters (see Section 4.2). The proposed cWGAN-GEP has only 3K parameters (see Table 1).
>
> We note that the generative performance of the LSTM baseline is much worse than the cWGAN-GEP in Fig. 4 where the MSE of cWGAN-GEP (10.4) is much lower than the LSTM baseline (15.89).
>
> **New Experiments Using Autoregressive LSTM:**: We expanded the ablation study to include an autoregressive LSTM as suggested. We carefully tuned the LSTM model (more tuning details can be found in Section 4.2) and eventually reached a LSTM based network containing 2 LSTM layers (with hidden state size of 15 and 10, respectively) and a fully connected layer (with size of 10). Then, the output at each time step is used to fit an autoregressive model to make final predictions. We note that the autoregressive LSTM has 4.3k parameters, which is more than the cWGAN-GEP. We repeat the experiment in Fig. 4 and calculate the mean squared error of the synthetic data generated by the autoregressive LSTM. The mean squared error of the autoregressive LSTM is 15.68 which is slightly better than the original LSTM baseline (15.89). However, it is still worse than the proposed cWGAN-GEP (10.4).
>
> **2. The Ensemble Approach Performs Worse Than The Proposed GenF:** Thanks for mentioning the interesting ensemble approach. As suggested, we repeated the experiment for predicting systolic blood pressure and trained two independent LSTM models using iterative and direct forecasting strategies, respectively. We note that both LSTM models are tuned via a validation set using grid search. The tuned LSTM model contains 2 LSTM layers (each with hidden state size of 10) and two fully connected layers (with size of 10 and 1, respectively). At each prediction horizon, we apply two ensemble approaches: averaging and weighted sum. For the averaging approach, we average the outputs from two LSTM models. For the weighted sum approach, we train a perceptron to calculate the weighted sum of the outputs from two LSTM models. The new results are shown below. It can be seen that the ensemble method performs worse than the proposed GenF.
>
> +++++++++++++++++++++++++++++++++++++++++++++++++++++\
> Prediction Horizon---------------t + 4------------t + 8--------------t + 12
> (GenF-1)-----------------------------114.7-----------144.5------------156.2\
> (GenF-2)-----------------------------108.5-----------141.7------------159.2\
> (GenF-3)-----------------------------101.5-----------133.6------------151.7\
> Ensemble (Averaging)-----------139.5-----------165.9------------201.5\
> Ensemble (Weighted Sum)-----128.7-----------151.6------------180.3\
> +++++++++++++++++++++++++++++++++++++++++++++++++++++
>
> **3. Larger Performance Gap for Longer Prediction Horizon:** We note that, in some plots, the performance gap between the LSTM baseline and the GenF (e.g., Fig. 3 (a)) does not change a lot, we think the main reason is that we only show the performance up to t+12. We expanded the experiment for predicting systolic blood pressure for a longer prediction horizon, namely t+30, t+60 and t+90. The MSE performance and Percentage Improvement are shown below. It can be clearly seen that the performance gap increases as the prediction horizon grows.
>
> ++++++++++++++++++++++++++++++++++++++++++++++++++++++++++++++\
> Prediction Horizon------------t + 12----------t + 30-----------t + 60----------t + 90\
> (GenF-1)--------------------------156.2-----------291.7-----------377.6----------457.6\
> (GenF-2)--------------------------159.2-----------262.9-----------358.3----------431.1\
> (GenF-3)--------------------------151.7-----------243.2-----------306.1----------398.3\
> LSTM (Direct Forecast)-------179.8-----------392.5-----------492.3----------654.4\
> LSTM (Iterative Forecast)----189.3-----------367.4-----------517.4--------- 688.5\
> Improvement (%) -------------18.5%-----------33.8%----------37.8%---------39.1%\
> ++++++++++++++++++++++++++++++++++++++++++++++++++++++++++++++
>
> We would like to thank Reviewer 1 again for bringing up several very important questions. We have addressed actionable suggestions and we hope all doubts are cleared. We will update our paper with the new results and we are glad to clarify if you have any further questions.

---

### Official Review · AnonReviewer2 · 2020-10-27
**Official Blind Review #2**

**Rating:** 5
**Confidence:** 3

**Review:**

This paper presents a framework for long-range forecasting of time series data using synthetic data. The whole pipeline contains three main components: cWGAN-GEP to predict/generate the following data based on observed ones; an LSTM predictor to forecast results based on real/generated data; and ITC to cluster data points by mutual information which forms training data for the first two components according to clustering results. The proposed framework is evaluated on three public datasets and results seem promising.

Pros:

1. This paper is well-structured and easy-to-follow.
2. The idea of utilizing synthetic data for improving the long-range forecasting of time series data is interesting.
3. The proposed method is widely evaluated on three datasets and the results are promising.

Cons:

My main concern towards this paper is that it lacks novelty on the technical side.

1. In cWGAN-GEP, adding MSE loss (mean squared error) for data generation (or augmentation) is a quite standard principle in the literature. It is hard to say this is novel.
2. Using LSTM for long-range data prediction is also widely used given LSTNet and TLSTM.
3. I think the main technical contribution of work is the Information Theoretic Clustering (ITC) algorithm.

Therefore, the contributions of this work are over-claimed by the authors in the introduction part and its technical contribution is minor. Overall, I think the current version of this paper cannot meet the bar of ICLR.

---

> ### Author Response · Authors · 2020-11-19
> **Response to Reviewer 2**
>
> We would like to thank you for your valuable time and concise comments. Please see our responses below.
>
> **1. On Novelty** : In summary, our work provides a novel forecasting strategy and a new neural network architecture (implemented via the ITC algorithm) that beats a diverse range of state-of-the-art benchmarks and classical approaches. Specifically, we think the novelty is three-fold:
>
> (i) **A Novel Forecasting Strategy**: Virtually all multi-step ahead forecasting algorithms are based on direct and/or iterative forecasting strategies. The proposed Generative Forecasting (GenF) provides a new forecasting strategy which balances the existing two forecasting strategies (direct and iterative forecasting strategies). To the best of our knowledge, our work is the first to do this.
>
> (ii) **A New Neural Network Architecture (cWGAN-GEP)**: We agree that using MSE as part of the loss function is well known. However, we note that incorporating the MSE term into cWGAN-GP for synthetic data generation is novel. In fact, in our ablation study (Section 4.4), we showed that adding this simple MSE term greatly improves generative performance. We believe our work is the first to do this.
>
> (iii) **The ITC Algorithm**: The ITC algorithm is designed to improve the performance of the basic GenF and we demonstrate that it works as expected (see Fig. 4). We note that most reviewers find the ITC algorithm to be a novel and important contribution.
>
> **2. On the use of LSTM**: We agree that the LSTM architecture has been widely used in forecasting. We would like to point out that our contribution is not the use of LSTM, but rather the hybrid forecasting strategy, the cWGAN-GEP and the ITC algorithm. We note that our approach outperforms modern forecasting benchmarks (such as LSTNet, TLSTM, and MLCNN). We compare our approach to these modern algorithms below.
>
> In LSTNet, the authors use LSTM together with the attention mechanism for long-range time series forecasting. In MLCNN, the authors use LSTM together with the autoregressive model to fuse forecasting information of different time steps. In our work, we use LSTM with a completely new design (the cWGAN-GEP) to implement a novel forecasting strategy (GenF). The proposed GenF has much fewer parameters (5.3k parameters) than all studied state-of-the-art methods (MLCNN: 15.9k parameters; LSTNet 19.2k parameters; TLSTM: 8.4k parameters), but beats all of them in terms of forecasting performance.
>
> We would like to thank Reviewer 2 for your valuable time. We hope all doubts are now cleared and we are glad to clarify if you have any further questions.

---

### Official Review · AnonReviewer4 · 2020-10-29
**Novel generative TS prediction, extensive experiments, moderate win**

**Rating:** 6
**Confidence:** 4

**Review:**

Summary:  This paper proposes a technique for long range time-series forecasting that leverages a generative model to extend the existing time series to train a time-series model to make the final prediction at horizon N.  The authors design a GAN, exploring an enhanced loss function, and provide a methodology by which to train it.  They evaluate the combined approach on a number of different time series, while comparing the technique to both traditional forecasting approaches as well as more recent RNN/CNN-based methods.  On those datasets, the technique meets or exceeds performance measured by MSE, in some cases improving MSE by 10% or more.

=== pros ===

+ Long range forecasts are indeed important, and the work appears to be a novel amalgam of recursive and direct forecasting with synthetic data generated from a GAN.  The paper is well organized and citations are extensive.

+ The evaluation compares against a range of traditional and current forecasting techniques, and for those experiments illustrates performance that almost always exceeds that of other approaches.

+ The ablation study indicates that the modifications to the traditional GAN architecture (extended loss function) and the use of mutual information to best train the GAN and predictive model provide benefits.

=== cons ===

- The paper chooses to use MSE to compare models.  This metric has some drawbacks; it is not at the scale of the data and it amplifies changes in error.   The paper claims 10% improvements in some time series, but this is 10% WRT MSE.  MSE inflates the perceived benefit.   Using MAE or RMSE would be more appropriate, though time series challenges even eschew those metrics in favor of symmetric MAPE (sMAPE) and MASE (see M4).  This raises the possibility that the concluding statement of "significantly outperform(ing)" other techniques may be a bit exaggerated.

- The evaluation investigates hourly and yearly time series.  It would be good to at least include a daily time series, and try to use a legitimately long-range forecasting horizon, like 30, 60, or 90 days.

- It's great to have the ablation study, but it is essentially anecdotal, as it only explores the performance for a single time series.

- Model comparisons are typically performed by time series cross validation (or in the case of hyperparameter searches, nested TS CV).   This would help to remove the possiblity of leakage when training on time series that might have correlations across time across sites (such as energy consumption or air quality).

=== suggestions / questions ===

I enjoyed this paper and its topic.  Multi-step ahead forecasts always force the practitioner to choose between direct, recursive, or seq2seq modeling approaches.   Evaluating a new technique that combines the two is a useful exploration.

It would be good to quantify what is meant by "long-range" forecasting.   Is it in steps or time?   As noted above, sometimes you have to forecast perhaps hundreds of steps into the future.  I'd probably call the TS forecasts here "medium-range", though this does not invalidate your results.

The motivation for ITC could be fleshed out -- it's a core contribution.   It would be good to motivate why a uniform random sampling of TS at the unit level does not lead to the best training data sets.   ITC ranks each unit time series by the sum of its MI score with all other units, and then creates Z same sized groups with similar scores.  Might explicitly say the sampling was proportional to J.   S3.4 claims that units with the same J scores tend to be dependent on each other.  What about the group with the lowest scores?

It is not clear whether you tune the hyperparameters of each direct forecasting model for each horizon.

Because you balance direct and recursive forecasting, a discussion comparing to RECTIFY (another approach that combines the two that you cite) might be useful in the related work section.  In addition, it would be good to mention that the Yoon 2019 NeurIPS paper trained predictive models on the generated time series; it might not have been "long range" but it's an important aspect of that work.

There are so many variations on your approach -- it would be great to say why you didn't evaluate them.  They include using the generative model for predictions (directly and recursively), or using an interative predictor instead of a direct predictor.  Similarly, why just generate 3 steps?

S3.4 You mention that ARIMA models multi-step prediction performance is equivalent between direct and recursive.  A direct N-step ahead model would have to use lags at least N days in the past.  Other work has found recursive outperforming direct (https://arxiv.org/pdf/1108.3259.pdf).   In our own work we have found that, for AR models, it depends on the time series itself.

==== nits ====

Please learn the difference between that and which.   Remove "in order to."

Abstract "for THE next few time steps"

S3.2P1 "improve classification results"
S3.2P2 "to generate synthetic"

---

> ### Author Response · Authors · 2020-11-19
> **Response to Reviewer 4 (Regarding the Suggestions / Questions)**
>
> **Regarding the suggestions / questions.**
>
> #1. To clarify, the ‘long-range’ is quantified in steps. As noted above, we have computed the performance for longer time horizons (up to t+90).
>
> #2. Regarding the ITC algorithm, our intuition is that the unit in the high score group tends to be more common as more units have high dependencies with it, and the unit in the low score group tends to be more rare as fewer units have dependencies with it. To select a representative subset, we should select both common and rare units. We plan to explore the ITC algorithm further in future work.
>
> #3. Yes, in our program, all forecasting models are tuned for each prediction horizon to ensure fair comparison.
>
> #4. Thanks for mentioning those interesting papers, we will add a discussion about the RECTIFY and the NeurIPS 2019 Paper.
>
> #5. The number of generative steps (L) depends on the quality of the synthetic data. In our experiments, we only show up to 3 time steps as our results suggest that generating more time steps is not beneficial to the forecasting performance (perhaps due to the error propagation effect). Lastly, we highlight that the parameter L can be treated as a hyper parameter and tuned via validation dataset.
>
> #6. In the experimental datasets, the difference between ARIMA using direct forecasting and iterative forecasting is marginal (up to t + 12). We have implemented ARIMA for longer horizons such as t + 30, t + 60 and t + 90. Our results for predicting systolic pressure suggest that the ARIMA with direct forecasting is slightly better. For other datasets, we are not sure for now, but we will explore them and update our paper accordingly.
>
> We would like to thank Reviewer 4 again for the kind words and useful review. Your suggestions have helped to improve our paper. We have conducted several new experiments and we hope that all doubts are now cleared. We note that, due to limited time, we only conducted some experiments for 5 runs and we will keep running the experiments and update the paper accordingly. We are glad to clarify if you have any further questions.

---

> ### Author Response · Authors · 2020-11-19
> **Response to Reviewer 4 (Regarding the cons)**
>
> We would like to sincerely thank you for your valuable time, constructive comments and suggestions. To address your comments about the cons, we have conducted some additional experiments as follows.
>
> **1. MAE for Predicting Systolic Blood Pressure for longer time horizons (for t+30, t+60 and t+90):**  We repeated the experiment for predicting systolic blood pressure for longer prediction horizons (i.e., t+30, t+60 and t+90). The results are shown below. In the table, we calculate the improvement (%) as compared to the best performing benchmark. From the table below, we observe that the improvement (in terms of MAE) becomes slightly smaller than our stated 10% improvement for t+4, t+8 and t+12, but it is interesting to note that the improvement for longer prediction horizon (i.e., t + 30, t + 60) tends to be larger, and this improvement is slightly higher than 10%.
>
> ++++++++++++++++++++++++++++++++++++++++++++++++++++++++++++++++++++++++++\
> Prediction Horizon---------t + 4--------- t + 8--------t + 12--------t + 30--------t + 60-------t + 90\
> (GenF-1)-----------------------5.87----------6.69---------7.88---------11.34--------13.18-------15.40
> (GenF-2)-----------------------5.57----------6.58---------8.09---------11.01--------12.85-------14.97\
> (GenF-3)-----------------------5.25----------6.21---------7.52---------10.58--------12.43--------14.36\
> MLCNN------------------------5.69----------6.65----------8.11--------11.89--------13.89--------16.21\
> TLSTM--------------------------5.91----------6.83----------8.28--------12.31--------15.09--------16.78\
> LSTNet-------------------------6.08----------6.92----------8.21---------12.08-------14.25--------16.53\
> Improvement----------------8.38%--------7.08%-------7.84%------12.38%------11.75%-----12.88%\
> ++++++++++++++++++++++++++++++++++++++++++++++++++++++++++++++++++++++++++
>
> **2. Ablation Study on More Patients:** The ablation study only explores the performance for a single time series (patient ID 23) so that we can have a closer look on the performance of cWGAN-GEP and other methods. We have evaluated other patients (e.g., patient ID 109, 890) and have observed similar performance trends. We will present the ablation study results for these additional patients in the appendix of the revised paper.
>
> **3. Population-Informed Cross Validation (PICV) and Time Series Cross Validation (TSCV):** We note that our cross validation (CV) method of random train-test split (also known as PICV) is based on the assumption of independence among units. First, we would like to highlight that the training set, validation set and test set are the same for all methods in each random split. We perform this by setting the random seed to ensure fair comparison. Therefore, no method gets extra benefits from the correlation even if there is a correlation across sites.
>
> However, we do see the reviewer’s point about data leakage. Hence, we repeated our experiments for predicting NO2 emission (the air quality dataset) using TSCV. Specifically, all datasets are split into training set (60%), validation set (20%) and test set (20%) in chronological order to ensure that there is no potential data leakage from using ‘future’ observations from the other units.  We tune all parameters using the validation set and compare the performance (mse) of using TSCV to PICV as follows.  Generally speaking, we find out that the performance of TSCV is roughly comparable to the performance of PICV. It is hard to conclude that the performance is affected by the correlation across sites, but the performance of the GenF still outperforms all benchmarks in the case of TSCV.
>
> ++++++++++++++++++++++++++++++++++++++++++++\
> Prediction Horizon-------t + 4-------t + 8-------t + 12
> -----------------------------PICV (Original)----------------------\
> (GenF-1)--------------------465.7------715.9------856.6\
> (GenF-2)--------------------409.3------669.4------814.6\
> MLCNN---------------------566.4------847.6-----1092.7\
> TLSTM-----------------------548.4------867.5------993.7\
> LSTNet----------------------518.4------827.6-------919.7\
> -----------------------------TSCV (New)--------------------------\
> (GenF-1)--------------------509.2------688.4------893.5\
> (GenF-2)--------------------433.8------651.2------848.3\
> MLCNN----------------------593.2------866.9-----993.5\
> TLSTM-----------------------603.5------819.3-----1023.5\
> LSTNet-----------------------557.4------790.3-----940.7\
> ++++++++++++++++++++++++++++++++++++++++++++

---

### Author Response · Authors · 2020-11-24
**Summary of changes to be made in the revised paper**

We would like to thank all reviewers for their valuable time and constructive comments. We have done various new experiments (4 in total) to address your comments and will improve the paper accordingly. We now summarize a list of changes to be made in the revised version:

**The following new results will be summarized in the revised paper:**

(i) Present forecasting performance using mean absolute error (MAE) and forecasting performance for longer prediction horizons (i.e., t + 30, t + 60, t + 90).

(ii) Include autoregressive LSTM in the ablation study and present ablation study for more patients (e.g., patient ID 109, 890).

(iii) Compare the forecasting performance of different cross validation (CV) methods. Specifically, we compare population-informed CV to time series CV.

(iv) Compare the proposed GenF to ensemble approaches (i.e., weighted sum and averaging) to justify the necessity of the two-stage approach.

**Regarding the paper writing:**

(i) Include an additional paragraph for complexity comparison. We would like to point out that the proposed GenF has much fewer parameters (5.3k parameters) than all studied state-of-the-art methods (MLCNN: 15.9k parameters; LSTNet 19.2k parameters; TLSTM: 8.4k parameters), but beats all of them in terms of forecasting performance. Furthermore, we will compare the model complexity of the cWGAN-GEP (3k parameters) to other generative models (e.g., the LSTM baseline with 3.5k parameters) to demonstrate the advantage of the cWGAN-GEP.

(ii) Address all suggestions/questions mentioned by Reviewer 4 (e.g., add a discussion about the RECTIFY and the NeurIPS 2019 Paper).

---

### Decision · Program_Chairs · 2021-01-07
**Final Decision**

**Decision:**

Reject

**Comment:**

This paper tackles the problem of long term time series forecasting. One challenge in long term forecasting is that often no sufficient date may be available. This paper proposes to use GANs to generate data that can be used to improve long-range forecasts.

While reviewers agree that this paper presents an interesting idea towards tackling the problem of long term time series forecasting and appreciate the effort authors put forward in addressing their concerns and comments during the response period, the reviewers believe that in the current form paper is not ready for the publication. In particular:

1. Reviewers find that overall technical contribution of this work is limited compared to other submissions.
2. Comparison of LSTM and cWGAN-GEP in the response is only on synthetic data, which does not address reviewers concerns.